# Examining recurrent hurricane exposure and psychiatric morbidity in Medicaid-insured pregnant populations

**Kelsey Herbst**[1], **Natasha P. Malmin**[2], **Sudeshna Paul**[3], **Trey Williamson**[1], **Margaret M. Sugg**[4], **Carl J. Schreck**[1], **Jennifer D. Runkle**[1] *

**1** North Carolina Institute for Climate Studies, North Carolina State University, Asheville, North Carolina, United States of America, **2** School of Public Health Department of Population Health Sciences, Georgia State University, Atlanta, Georgia, United States of America, **3** Nell Hodgson School of Nursing, Emory University, Atlanta, Georgia, United States of America, **4** Department of Geography and Planning, Appalachian State University, Boone, North Carolina, United States of America

* jrrunkle@ncsu.edu

**Data Availability Statement:** Data used in this study involve human subjects research participant data and are considered sensitive data. Data on pregnant persons contain private identifiable

## Abstract

The wide-ranging mental health consequences of a major hurricane have been widely documented, but important gaps remain in understanding the mental health burden of recurrent exposure to multiple hurricanes. The objective of this study was to examine the mental health burden in at-risk pregnant populations recurrently exposed to major hurricanes Matthew (2016), Florence (2018), Michael (2018), and Dorian (2019). Daily emergency department (ED) admissions were obtained on pregnant Medicaid beneficiaries for incident maternal disorders of pregnancy (MDP), perinatal mood and anxiety disorders (PMAD), severe mental illness (SMI), and substance use (SUD). County-level hurricane exposure was derived from a novel meteorologic-based Hurricane Insurance Protection- Wind Index (HIP-WI) metric. A difference-in-difference analysis assessed county-level changes in ED-related visits for psychiatric morbidity in pregnant populations following single hurricane events and a marginal Generalized Estimating Equation model assessed the cumulative impact of recurrent county-level hurricane exposure. A total of 258,157 (59.0%) pregnant cases were exposed to no storms, 113,157 (25.8%) were exposed to one storm, and 66,407 (15.2%) were exposed to two or more storms. Results showed higher risks for MDP after Matthew (RR: 1.83, 95%CI: 1.53, 2.18) and after Florence (RR: 1.09, 95%CI: 0.99, 1.19); higher risk of SMI (RR: 1.46, 95%CI: 1.11, 1.91) and PMAD (RR: 1.52, 95%CI: 1.32, 1.74) after Matthew. Compared to no storm exposure, exposure to two or more storms was associated with a higher risk of MDP (RR: 1.58, 95% CI [1.47,1.63]); PMAD (RR: 1.51, 95% CI [1.44, 1.59]); and SMI (RR: 1.34, 95% CI [1.23, 1.47]). Access to maternity care services, urbanity, and economic and residential segregation were important effect modifiers. Our population-based ecological study demonstrated that cumulative hurricane exposure confers an increased risk for psychological morbidity in pregnant Medicaid beneficiaries, particularly for mood and anxiety disorders, incident mental disorders, and severe mental illness for a Southern state outside of the U.S. Gulf Coast.

information and sharing is prohibited as stipulated by the data use agreement. For persons interested in obtaining these data, please contact: Cecil G. Sheps Center for Health Services Research The University of North Carolina at Chapel Hill CB# 7590 725 Martin Luther King Jr. Blvd. Chapel Hill, NC 27599-7590 contact@schsr.unc.edu P: 919-966-5011.

**Funding:** This work was supported by the National Institutes of Health (1R03ES031228-01A1 to JR, MS) and the National Oceanic and Atmospheric Administration (NA21OAR4310312 to JR, NM, CR; NA19NES4320002 to JR, KH, CS, TW). The funders had no role in study design, data collection and analysis, decision to publish, or preparation of the manuscript.

**Competing interests:** The authors have declared that no competing interests exist.

## Introduction

New research demonstrates that the damaging effects of tropical hurricanes (e.g., extreme flooding and wind speeds) are being exacerbated by the compounding effects of sea level rise, increasing sea surface temperatures, and higher precipitation variability brought on by a changing climate [1, 2]. As a result, recent tropical cyclones in the U.S. have been wetter, rapidly intensifying, and slower moving, with significant potential to adversely impact the health and well-being of a larger portion of the U.S. population than ever before [3]. Prior research has linked hurricane exposure and storm severity to higher rates of psychological disorders like anxiety, post-traumatic stress disorder (PTSD), major depressive disorders, suicidality, and psychological distress induced substance misuse in directly impacted communities [4–10]. Yet, disasters do not impact all populations equally, even within a geographical area, and women may have a stronger mental health response to disasters than men [11–13]. Vulnerable subgroups, like pregnant populations, are more likely to report significant physical and psychological health complications in the aftermath of these severe tropical storms [14]. Climate-intensified hurricanes can adversely impact pregnancy health through direct exposure and indirectly through changes in the physical (e.g., inland flooding, indoor mold in homes) and social (e.g., housing/water/food insecurity, job loss, interpersonal violence) environment [15]. Disruptions to housing, transportation, healthcare services, and social supports complicated by economic hardships (e.g., damage to home, loss of job) in the aftermath of a major hurricane have been associated with increases in incident and long-term morbidity for women and their children [16].

Maternal exposure to major hurricanes has been linked to adverse health outcomes during pregnancy. Pregnant populations who experienced Hurricane Irma (Category 5) and Hurricane Maria (Category 4) were at a higher risk for pregnancy-related complications [17, 18], including early onset labor and unplanned cesarean birth [17, 19, 20], severe maternal morbidity, and hypertensive disorders [17]. Following the landfall of Hurricane Sandy (Category 3) in New York, a significant increase in total pregnancy complication visits persisted up to three months post-storm [18]. The risk of severe maternal morbidity and hypertensive disorders of pregnancy were significantly higher in low-income women impacted by Hurricane Harvey [17]. Pregnant populations residing in Puerto Rico and who were impacted by Hurricane Irma and Maria reported traumatic stress and depressive symptoms, posttraumatic stress symptoms, and anxiety [19], particularly among women who experienced greater hurricane exposure and who had poor social support [21]. While post-traumatic stress and psychological distress are commonly reported in directly impacted communities, few studies have examined psychiatric morbidity in pregnant populations following single hurricane events, and even fewer have examined the risk of psychiatric disorders following multiple disaster exposure to hurricanes.

In the last decade, the public health impacts of multiple disaster exposure, including natural, technological, climate-driven, and social disasters, on individuals and communities have emerged as a research priority [22]. In the Southeastern U.S., the Atlantic hurricane season has been especially active due to a *La Niña* effect, ushering in 4 major storms, including Hurricanes Matthew (2016), Florence (2018), Michael (2018), and Dorian (2019). Communities in the Southeast have been in constant recovery from these large storms, but no studies have examined the excess mental health burden in at-risk pregnant populations in communities following repeated exposure to major hurricanes. Borrowing from Leppold, conceptually multiple disaster events have been characterized as cascading disasters (i.e., events that generate unexpected secondary events with significant impact), compound disasters (i.e., two or more hazards that occur simultaneously or in succession), or recurrent disasters (i.e., recurrence of a

single hazard in the same geographic region) [22]. Several methodological gaps remain in quantifying the health impacts of multiple disasters, including a lack of a general consensus on the best practices for measuring (direct or indirect) exposure to multiple or recurring exposures, defining the temporal period(s) or return interval between events that adequately reflect repeat or recurrent exposure, or identifying methodological approaches that capture changes from baseline across multiple events that occurred over space and time [22].

For this analysis, we focus on the mental health impacts of recurrent exposures in the prenatal period to multiple tropical cyclones. In this paper, we conceptualize recurrent exposure as the repeat occurrence of a climate-induced hurricane in the same geographic location, recurring typically on an annual basis. The objective of this study was to examine the mental health burden in pregnant populations residing in North Carolina communities recurrently exposed to multiple tropical cyclones, including Matthew (2016), Florence (2018), Michael (2018), and Dorian (2019). Relying on county-level data from a novel meteorologic-based hurricane insurance program, we will employ a new metric in environmental epidemiology for measuring population-based hurricane exposure. We will first examine the impact of single storms using a difference-in-difference design-based approach to causal inference and then examine recurring tropical cyclone exposure and maternal mental health burden using a population-based marginal model. We will identify community-level characteristics (e.g., rural versus urban, index concentration of extremes) associated with increased risk of ED visits for mental health disorders during pregnancy. We hypothesized that pregnant persons in counties recurrently exposed to multiple storms would shoulder a higher excess burden of psychiatric morbidity than women in counties who did not experience a hurricane. We also hypothesized that women residing in rural locations or with low maternal care access would have a higher mental health burden in response to recurrent disasters. The majority of the health impact literature to-date emphasizes disaster resilience and recovery efforts from a single-event framing of a disaster, but viewing the health of impacted populations from disasters as complex and recurring events brings forth a more nuanced view from which to derive targeted interventions and more responsive disaster recovery policies.

## Materials and methods

### Maternal outcomes

Daily emergency department (ED) visits for pregnant persons between the ages of 18 and 44 years were obtained only on Medicaid beneficiaries from the UNC Sheps Center [23]. Pregnant persons with a spontaneous or elective abortion or who showed up to the ED for childbirth were excluded. ED visits for pregnant cases were identified first by relying on diagnosis codes relating to maternal care for pregnancy-related conditions (O26) and conditions that develop as a result of pregnancy (O99). We then used the International Classification of Diseases, tenth revision (ICD-10) diagnosis to code the following binary maternal mental health outcomes (yes, no) in this analysis (S1 Table): (1) a new maternal mental health disorder (MDP); (2) perinatal mood or anxiety disorder (PMAD), is an existing depressive, anxiety, or stress disorder that occurred before pregnancy; (3) severe mental illness (SMI), including existing severe psychotic disorders like mania, schizophrenia, and bipolar disorder before pregnancy; and (4) substance use disorder complicating pregnancy. Week of gestation was coded using Z3A.XX. For this population-based ecological study, ED visits were accessed from October 1, 2015, to September 1, 2020. In addition to an individual's past and current health or conditions coded by specific diagnosis codes, ED admissions also included individual-level data on maternal age (categorized as 18–24, 25–34, and ≥35 years), race/ethnicity (White non-Hispanic, Black non-Hispanic, Hispanic, and Other including Asian, American Indian, and

Pacific Islander), date of admission, and county of residence; while these data are de-identified there is some risk of reidentification in which the de-identified data could potentially be linked back to the identity of a patient.

## Tropical storm exposure assessment

Public hurricane- and tropical storm-related crop insurance coverage data was obtained from the Hurricane Insurance Protection—Wind Index (HIP-WI) produced by the United States Department of Agriculture (USDA) [24]. The HIP-WI is a supplemental crop insurance program intended to provide coverage to agricultural workers in counties exposed to sustained tropical storm-strength winds, limited to named storms [25]. Wind extent is determined by meteorological variables derived from the International Best Track Archive for Climate Stewardship (IBTrACS) data from the National Oceanic and Atmospheric Administration's National Hurricane Center. To issue a county loss trigger: a) a hurricane must have maximum sustained surface winds of 64 knots (74 mph) or greater, or b) a tropical storm event must have been characterized by a sustained surface wind speed of 34 knots (39 mph) and at least 6 inches of total precipitation accumulating over four consecutive days (i.e., one day before the storm, the day of the storm, and two days after the storm hits). Individual convex hulls are computed based on the initial and final tropical storm points, the estimated center point, along their corresponding oval buffers. Convex hulls that overlap with county boundaries defined by the Census Bureau indicate counties exposed to hurricanes and determine county loss triggers for reimbursement [26]. To approximate storm exposure for North Carolina counties, we utilized published county loss triggers to distinguish between exposed and unexposed counties. Multiple factors were used to gauge storm exposure, including storm assigned a name, diameter in nautical miles, total rainfall, and wind speed [26]. To-date, no studies have used this meteorologic-based metric to characterize population-level exposure to a tropical storm. Counties that experienced loss triggers as identified by the HIP-WI for a given storm were considered 'exposed' and assigned to the treatment group. Counties that did not experience loss triggers disaster for a given storm were assigned to the unexposed or control group.

## Covariates

Maternal mental health during disaster recovery has been shown to differ based on residence in a rural compared to an urban location [27, 28]. The USDA Rural-Urban Community Area (RUCA) codes by county of residence were used to examine mental health disparities for urban (codes 1–3), suburban (codes 4–6), and rural (codes 7–10) communities. One important factor that explains rural-urban differences in psychiatric morbidity is access to care, and limited research has shown that access to maternal care is often disrupted in the aftermath of large-scale hurricanes [20]. In NC, 21% of counties have been designated as maternity care deserts, with the vast majority of those counties being in underserved and rural communities [29]. To better understand the mental health of pregnant persons and access to care, we included a categorical variable to measure residence in a maternity care desert (0 = full access to maternity care, 1 = low access, and 2 = maternity care desert) [29]. Lastly, poverty and racism may elevate risk for depression or another mental health disorder [30, 31] and living in disadvantaged or dangerous neighborhoods may interact with negative life experiences to increase susceptibility to a mental health condition [32]. We operationalized residence in a racially or economically segregated neighborhood using the Index of the Concentration of Extremes (ICE) to simultaneously capture extremes of privilege and deprivation. ICE residential segregation or ICE economic segregation is a continuous variable where -1 indicates

extremes of low-income or majority black and 1 indicates extremes of high-income or majority white communities, respectively. Tertiles were used to categorize each ICE metric (i.e., T1: low-income, T2: mixed income, T3: high-income for ICE economic segregation or T1: majority Black, T2: mixed race, T3: majority White communities for ICE residential segregation).

## Statistical analysis

Pregnant patient demographic data on maternal age and race/ethnicity, as well as community-level factors, including maternal care desert, urbanicity, and ICE metrics were compared between pregnant cases for all storms combined using chi-square tests to compare differences between categorical demographic variables.

**Difference-in-difference.** We implemented a quasi-experimental longitudinal study design using the difference-in-difference (DID) analytic approach to examine differences in psychiatric morbidity for pregnant cases residing in hurricane impacted and control counties for each of the four storms separately. DID is a commonly used approach for causal inference and has been used to examine flood impacts on pregnancy health and birth outcomes [33, 34]. One advantage to the pre-post design includes the time-ordering of events, allowing for the subtraction of background trends observed in the control group from the change in the outcome due to hurricane exposure in the intervention group; thereby enhancing the interpretation of model results through a causal lens [35]. The DID method is an econometric approach that helps answer the question of what would have happened to maternal mental disorders if the intervention (i.e., hurricanes) had not taken place. The DID is a quasi-experimental methodology that estimates the true impact of the intervention (e.g., hurricane). In this study, we analyzed repeated cross-sectional county-level data that followed different groups of pregnant populations residing in hurricane-impacted counties and control or non-impacted counties one year before and one year after each individual hurricane. Each county served as its own control and consists of repeated observations of maternal mental health-related ED visits for individual counties; whereby the assumption is that county-level psychiatric morbidity in a given county before the hurricane serves as a counterfactual for what county-level psychiatric morbidity in pregnant populations would have been in the absence of a given hurricane event. In the DID methodology, the impacted group is exposed to a hurricane (i.e., intervention), and the control group is not exposed to the intervention. The DID design then compares the changes within each group (intervention versus control) between the pre-treatment (i.e., the first difference) and post-treatment (i.e., the second difference) periods. A DID estimator is then computed to compare the differences between the intervention and control group post-hurricane (second difference), while accounting for the existing differences between intervention and control groups pre-event (first difference). Another key advantage of the DID or quasi-experimental design is that the assignment of treatment or control group simulates that of a randomized control trial, and with the addition of the propensity score weight, controls for unobserved and observed confounders that are constant over time, resulting in balanced treatment and control groups.

The DID method was used to examine the relationship between county-level storm exposure and the risk of psychiatric morbidity during pregnancy before and after 4 major storms impacting North Carolina. Maternal mental health outcomes in pregnant cases for each of the 100 North Carolina counties based on storm exposure (exposed vs control) were aggregated into the four separate one year pre- and one year post-hurricane periods including: (1) *Matthew*: October 8, 2015—October 8, 2017 (pre-storm: October 8, 2015—October 7, 2016, post-storm: October 8, 2016—October 8, 2017); (2) *Florence*: September 12, 2017—September 12, 2019 (pre-storm: September 12, 2017—September 11, 2018, post-storm: September 12, 2018—

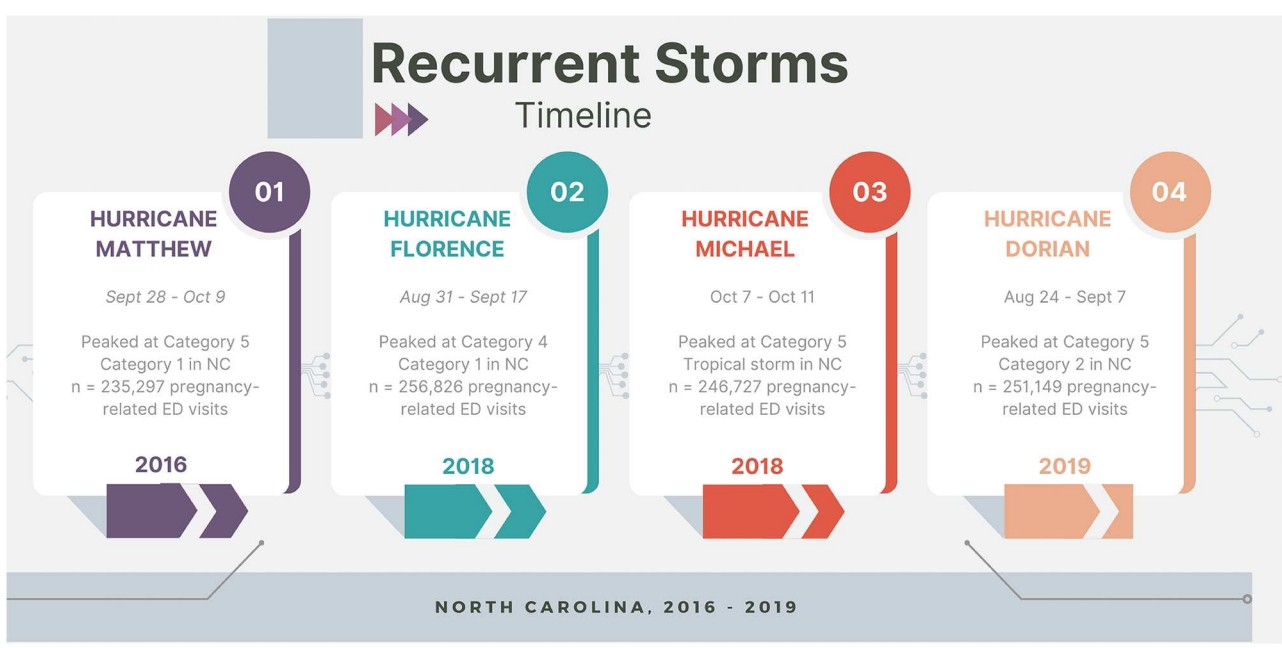

**Fig 1. Timeline for multiple hurricane events in North Carolina, 2016 to 2019.**

September 12, 2019); (3) *Michael*: October 10, 2017—October 10, 2019 (pre-storm: October 10, 2017—October 9, 2018, post-storm: October 10, 2018—October 10, 2019); (4) *Dorian*: September 1, 2018—September 1, 2020 (pre-storm: September 1, 2018—August 31, 2019, post-storm: September 1, 2019—September 1, 2020) (Fig 1). For each storm, only controls and treatment communities were selected for that storm. The parallel trends assumption held for each storm-impact model. Difference 1 was composed of (1) 'exposed' pregnant women living in a hurricane-impacted county (post-storm) and (2) 'control group 1' pregnant women living in a non-impacted county (post-storm) and difference 2 as 'control group 2' pregnant women living in a hurricane-impacted county (pre-storm) and 'control group 3' pregnant women living in a non-impacted county (pre-storm). During these designated periods, counties that experienced loss triggers identified by the HIP-WI for a given storm were considered 'exposed' and assigned to the treatment group. Counties that do not experience loss triggers for a given storm were assigned to the 'control group'.

Difference-in-differences analyses were conducted using a log-Poisson generalized linear model with interaction terms on the log scale. Risk estimates of mental health outcomes prior to and after a hurricane event were derived from ICD-10 claims codes. The average change in mental health impacts over a hurricane period were calculated separately among the exposed and unexposed groups. The mean difference between the two coefficients represents the causal effect of the hurricane event on each specified mental health outcome separately. Propensity score weighting was included in each DID model to account for any case-mix differences at baseline between the exposed and unexposed groups based on age and race/ethnicity across individual storm events. Propensity score weights were derived by modeling the probability of being in the hurricane-impacted group by including participant age and racial-ethnic status and were generated using Proc PSMATCH [35].

We conducted difference-in-difference analysis separately for each storm using PROC GENMOD and ESTIMATE statements in SAS version 9.4. Models then separately estimated

the risk ratios (RR) associated with each distinct maternal mental health outcome (e.g., PMAD, MDP, SMI, and SUDP) among exposed and unexposed individuals before and after each hurricane event [36].

**Cumulative impact model.** To study the cumulative impact of recurrent storm exposure on maternal mental health outcomes, we created a cumulative exposure variable by summing over the dichotomous hurricane exposures at the county level. For this population-based ecological analysis, we operationalized recurrent storm exposure as county-level experience of multiple hurricane events (e.g., Matthew, Florence, Michael, or Dorian) between Jan 1, 2016, to Dec 31, 2019, the study period for which all storms occurred. Counties were flagged as being "exposed" to a given storm if a loss trigger was flagged as identified by the storm-specific HIP-WI, and "control" counties were assigned based on the absence of a loss trigger. Possible values of the cumulative exposure variable ranged between 0 and 4. The final recurrent storm variable consisted of the following categories: 0 = no storm exposure, 1 = exposure to one storm, and 2 = exposure to two or more storms. We opted to categorize recurrent storm exposure in this way because there were so few individuals who experienced two recurrent storms only. Therefore, to ensure an adequate sample size, we combined a total of two or three storms experienced into the '2 or more storms' category. No counties experienced four consecutive storms. To account for the clustering of pregnant Medicaid beneficiaries within counties, we used generalized estimating equations (GEE) regression models. The marginal GEE models provide population-averaged effect estimates while accounting for dependency between individuals within the same county. Specifically, the correlation is accounted for by robust estimation of variance of the regression coefficients.

Separate modified Poisson regression models with the link function specified to log, a quasi-likelihood model, were fitted for each binary maternal mental health outcome and included the fixed effects of cumulative exposure and individual and community-level covariates [37]. The modified or robust Poisson model is the preferred population model-based approach for estimating risk ratios, particularly when models were misspecified [38, 39]. *Model 1* was the crude model. *Model 2* adjusted for individual-level factors of the pregnant person, including racial identity (non-Hispanic Black, non-Hispanic White, Hispanic, and Other race including American Indian, Asian, and mixed race), maternal age group (18–24 years, 25–34 years, 35 years+) and Model 3 adjusted for community-level covariates including maternal access to care (full access, low access, maternal care desert), urbanicity (urban, suburban, rural), and tertiles of residential segregation (T1: majority black, T2: mixed race, T3: majority white) and economic segregation (T1: majority low-income, T2: mixed income, T3: majority high-income). Final *Model 4* included significant individual and community-level covariates (p < .01). We also examined the differential storm effect for the following effect modifiers: maternal care access, urbanity, maternal residence in a racially segregated county, and maternal residence in an economically segregated county.

Proc Genmod in SAS (Version 9.4) was used to conduct the GEE analyses for the cumulative impact models [36]. Given the low prevalence of each maternal mental disorder (< 6% of the population), the odds ratio approximates the risk ratio, and crude and adjusted risk ratio (RR) with 95% confidence intervals were generated for each model. Pairwise comparisons of the different cumulative exposure classifications were computed. Interaction terms, one at a time, were added to assess effect modification by maternal care access, urbanity, and maternal residence in a racially or economically segregated county and used a threshold of p-value <0.10 to assess the significance of each interaction term (i.e., p-EM). Access to the Sheps Center ED data is restricted by a data use agreement, given the potential for identification of patients, and this study involving secondary analysis was determined exempt to the Institutional Review Board at North Carolina State University (protocol number: 24297). Any

identifiable information has been anonymized in the manuscript to protect the privacy of Medicaid patients in our sample.

## Results

During the entire study period, 258,157 (59.0%) pregnant cases were exposed to no storms, 113,157 (25.8%) were exposed to one storm, and 66,407 (15.2%) were exposed to two or more storms (Table 1). A majority of mothers were less than 35 years of age (93.2%), were either non-Hispanic Black (50.8%) or non-Hispanic White (38.4%), and living in an urban area (80.3%). A larger proportion of non-Hispanic White women (41.9%) resided in counties who were not exposed to a hurricane compared to those exposed to two or more storms (38.6%); conversely, a larger proportion of non-Hispanic Black women resided in counties that experienced two or more storms (52.0%) compared to Black women in counties with no storm experience (47.6%). Further, more women impacted by two or more storms resided in a maternal care desert (4.6%) or community with low access to maternal care (26.7%), or a suburban area (40.9%) and were characterized by a higher prevalence of PMAD (7.8%), SMI (2.2%), or MDP (6.0%). Of the 238,901 pregnant Medicaid beneficiaries included in this analysis, 60,119 (14.1%), 90,069 (37.7%), 5,158 (2.2%), and 40,466 (16.9%) were exposed to individual hurricanes Matthew, Florence, Michael, and Dorian, respectively (S2 Table).

### DID results

Table 2 shows the marginal effects and 95% confidence interval (CI) for the DID estimator with propensity score weighting comparing the county-level changes in maternal mental health during the one-year pre- and one-year post-periods for each hurricane event using the HIP-WI metric. Overall, there were statistically significant differences in mental health-related ED visits between the post-storm compared to pre-storm periods, including higher risks for newly diagnosed maternal mental disorders of pregnancy (MDP) after Matthew (RR: 1.83, 95%CI: 1.53, 2.18) and after Florence (RR: 1.09, 95%CI: 0.99, 1.19); higher risk of severe mental illness (SMI) (RR: 1.46, 95%CI: 1.11, 1.91) and perinatal mood disorders (PMAD) (RR: 1.52, 95%CI: 1.32, 1.74) post-Matthew. Conversely, results showed a statistically significant reduction in ED-visits among pregnant Medicaid beneficiaries for MDP (RR: 0.67, 95%CI: 0.69, 0.76), PMAD (RR: 0.68, 95%CI: 0.60, 0.76), and SMI (RR: 0.70, 95%CI: 0.56, 0.88) in response to Dorian.

### Recurrent multivariate analyses

Results for the marginal GEE model examining recurrent hurricane exposure and maternal mental disorders can be found in Table 3. Overall, exposure to two or more storms, relying on the meteorological-based HIP-WI storm impact metric, was associated with a greater risk for MDP, PMAD, and SMI in pregnant Medicaid beneficiaries compared to Medicaid beneficiaries residing in counties not impacted by a hurricane. Compared to no storm exposure, after adjusting for individual and community-level covariates, exposure to two or more storms was associated with a higher risk of MDP (RR: 1.58, 95% CI [1.47, 1.63]); PMAD (RR: 1.51, 95% CI [1.44, 1.59]); and SMI (RR: 1.34, 95% CI [1.23, 1.47]). Interestingly, exposure to one storm compared to no storm experience was associated with a higher risk of SUBP (RR: 1.10, 95% CI [1.01, 1.19]).

Access to maternity care services, urbanity, and economic and residential segregation were important effect modifiers (Table 4). Pregnant Medicaid beneficiaries with full access and low access to maternal care services were at a higher risk of an emergency visit for MDP or PMAD following two or more storms compared to a reduced risk for women in maternity care

**Table 1. Characteristics of eligible pregnant Medicaid beneficiaries, North Carolina, 2015–2020.**

| Maternal Characteristics | Total | 0 Storms | 1 Storm | ≥ 2 Storms |
|---|---|---|---|---|
| | n (%) | n (%) | n (%) | n (%) |
| *Age* | | | | |
| 18–24 | 114285 (47.84) | 67301 (47.77) | 28252 (47.00) | 18732 (49.43) |
| 25–34 | 108323 (45.34) | 64091 (45.49) | 27529 (45.80) | 16703 (44.07) |
| ≥35 | 16293 (6.82) | 9504 (6.75) | 4325 (7.20) | 2464 (6.50) |
| *Race & Ethnicity* | | | | |
| White non-Hispanic | 89469 (38.36) | 58220 (41.88) | 17693 (29.94) | 13556 (38.60) |
| Black non-Hispanic | 118485 (50.81) | 66108 (47.56) | 34128 (57.76) | 18249 (51.96) |
| Hispanic | 17642 (7.56) | 10318 (7.42) | 5181 (8.77) | 2143 (6.10) |
| Other | 7616 (3.27) | 4359 (3.14) | 2083 (3.53) | 1174 (3.34) |
| *Geography* | | | | |
| Urban | 191788 (80.28) | 116934 (82.99) | 52920 (88.04) | 21934 (57.87) |
| Suburban | 37693 (15.78) | 17029 (12.09) | 5174 (8.61) | 15490 (40.87) |
| Rural | 9420 (3.94) | 6933 (4.92) | 2012 (3.35) | 475 (1.25) |
| *Access to Maternity Services* | | | | |
| Full Access to Care | 193376 (80.94) | 118709 (84.25) | 48620 (80.89) | 26047 (68.73) |
| Low Access to Care | 37137 (15.54) | 17370 (12.33) | 9663 (16.08) | 10104 (26.66) |
| Maternity Care Desert | 8388 (3.51) | 4817 (3.42) | 1823 (3.03) | 1748 (4.61) |
| *ICE: Economic Segregation* | | | | |
| T1: Low-income | 89612 (37.51) | 41061 (29.14) | 26340 (43.82) | 22211 (58.61) |
| T2: Moderate income | 76791 (32.14) | 58680 (41.65) | 7432 (12.36) | 10679 (28.18) |
| T3: High-income | 72498 (30.35) | 41155 (29.21) | 26334 (43.81) | 5009 (13.22) |
| *ICE: Residential Segregation* | | | | |
| T1: Majority Black | 85595 (35.83) | 55164 (39.15) | 22101 (36.77) | 8330 (21.98) |
| T2: Mixed | 68885 (28.83) | 25255 (17.92) | 27596 (45.91) | 16034 (42.31) |
| T3: Majority White | 84421 (35.34) | 60477 (42.92) | 10409 (17.32) | 13535 (35.71) |
| *Mental Health Outcomes* | | | | |
| MDP | 11268 (4.72) | 6581 (4.67) | 2402 (4.00) | 2285 (6.03) |
| SMI | 4219 (1.77) | 2448 (1.74) | 932 (1.55) | 839 (2.21) |
| PMAD | 14348 (6.01) | 8390 (5.95) | 3002 (4.99) | 2956 (7.80) |
| SUDP | 4555 (1.91) | 2820 (2.00) | 1066 (1.77) | 669 (1.77) |

Frequency missing: Race (n = 2938), Ethnicity (n = 3262).

MDP: Maternal mental disorder during pregnancy; PMAD: Perinatal mood or anxiety disorder: SMI: Severe mental disorder: SUBP: Substance use disorder complicating pregnancy.

**Table 2. Marginal effects and 95% confidence intervals (CI) for DID analysis assessing county-level changes in maternal mental health for exposed compared to control counties during the one year before and after each hurricane event, North Carolina 2015–2020.**

| Hip-WI | Matthew (2016) | | Florence (2018) | | Michael (2018) | | Dorian (2019) | |
|---|---|---|---|---|---|---|---|---|
| DID Estimator | RR | [95% CI] | RR | [95% CI] | RR | [95% CI] | RR | [95% CI] |
| MDP | 1.83 | [1.53, 2.18] | 1.09 | [0.99, 1.19] | 1.15 | [0.88, 1.51] | 0.67 | [0.59, 0.76] |
| SMI | 1.46 | [1.11, 1.91] | 0.89 | [0.76, 1.05] | 1.23 | [0.72, 2.11] | 0.70 | [0.56, 0.88] |
| PMAD | 1.52 | [1.32, 1.74] | 1.04 | [0.96, 1.14] | 1.12 | [0.86, 1.46] | 0.68 | [0.60, 0.76] |
| SUDP | 0.99 | [0.75, 1.30] | 1.11 | [0.96, 1.29] | 0.64 | [0.37, 1.11] | 1.16 | [0.93, 1.45] |

MDP = Maternal Disorders of Pregnancy; SMI = Severe Mental Illness; PMAD = Perinatal mood or anxiety disorder; SUDP = Substance use disorder complicating pregnancy. HIP-WI = Health Insurance Protection—Wind Index. RR = Risk Ratio.

**Table 3. Cumulative impact model output for recurrent storm impacts using the HIP-WI metric on maternal mental disorders, North Carolina (2016–2019).**

| Model | Outcome | RR (95% CI) | | |
|---|---|---|---|---|
| | | **0 Storms** | **1 Storm** | **≥2 Storms** |
| **Model 1:** Crude Model | MDP | 1 (referent) | 0.85 (0.81, 0.89) | 1.31 (1.25, 1.38) |
| | PMAD | 1 (referent) | 0.83 (0.80, 0.87) | 1.34 (1.28, 1.40) |
| | SMI | 1 (referent) | 0.89 (0.83, 0.96) | 1.28 (1.18, 1.39) |
| | SUDP | 1 (referent) | 0.88 (0.82, 0.95) | 0.88 (0.81, 0.96) |
| **Model 2:** Adjusted for individual-level factors | MDP | 1 (referent) | 0.92 (0.88, 0.97) | 1.41 (1.34, 1.49) |
| | PMAD | 1 (referent) | 0.92 (0.88, 0.96) | 1.45 (1.39, 1.52) |
| | SMI | 1 (referent) | 0.97 (0.90, 1.05) | 1.35 (1.24, 1.46) |
| | SUDP | 1 (referent) | 0.96 (0.90, 1.04) | 0.83 (0.76, 0.91) |
| **Model 3:** Adjusted for community characteristics | MDP | 1 (referent) | 1.03 (0.97, 1.08) | 1.48 (1.41, 1.57) |
| | PMAD | 1 (referent) | 0.95 (0.91, 1.00) | 1.42 (1.35, 1.49 |
| | SMI | 1 (referent) | 0.97 (0.90, 1.06) | 1.31 (1.19, 1.42) |
| | SUDP | 1 (referent) | 1.10 (1.01, 1.18) | 0.94 (0.86, 1.03) |
| **Model 4:** Combined model | MDP | 1 (referent) | 1.03 (0.98, 1.09) | 1.58 (1.49, 1.66) |
| | PMAD | 1 (referent) | 0.97 (0.93, 1.02) | 1.51 (1.44, 1.59) |
| | SMI | 1 (referent) | 0.99 (0.92, 1.08) | 1.34 (1.23, 1.47) |
| | SUDP | 1 (referent) | 1.10 (1.01, 1.19) | 0.93 (0.85, 1.02) |

MDP: Maternal mental disorder during pregnancy; PMAD: Perinatal mood or anxiety disorder: SMI: Severe mental disorder: SUDP: Substance use disorder complicating pregnancy. RR = risk ratio. CI = Confidence Interval.

*Model 2*: Adjust for individual factors (i.e., racial/ethnic identity, maternal age); *Model 3*: Adjusted for community characteristics (i.e., maternal care deserts, urban/rural status, ICE: economic segregation, and ICE: residential segregation); *Model 4*: Adjust for individual and community-level characteristics (i.e., racial/ethnic identity, maternal age, access to care, urban/rural status, ICE: economic segregation, and ICE: residential segregation)

deserts. Women residing in counties with 2+ storms and low access to care had 2.6 times higher risk for an ED visit for MDP (95%CI: 2.36, 2.93) compared to the no storm exposure group. Similarly for women in low access to care communities, we observed a twofold higher risk of SMI following two or more storms compared to no storms (RR: 2.03, 95%CI: 1.72, 2.39). Compared to no storm exposure, pregnant Medicaid beneficiaries in urban areas were much more likely to present to the ED for MDP, PMAD, or SMI following two or more storms. Contrary to what was expected, results showed a much higher risk of ED visits for psychiatric morbidity for pregnant populations residing in mixed income, majority high-income, or majority white communities. Similarly, we observed a stepwise increase in risks of MDP, PMAD, and SMI for pregnant Medicaid beneficiaries in majority White communities impacted by 2 or more storms compared to no storms. Yet in Majority Black communities, we observed a significantly lower risk of maternal ED visits for psychiatric morbidity in communities weathering 2 or more storms compared to those with no storm exposure. Lastly, a higher risk of an ED visit for substance use during pregnancy occurred in majority-Black communities experiencing two or more storms and in communities with low access to maternal care following one storm.

## Discussion

This study examined the impact of recurrent storm exposure on pregnant Medicaid populations presenting with emergency psychiatric disorders. In general, we observed a larger cumulative impact on mental health during pregnancy following exposure to multiple and recurring storms compared to no storm exposure or exposure to a single storm. Results examining

**Table 4. Adjusted risk ratios (aRR)[a] and 95% confidence intervals (95%CI) from cumulative impact model demonstrating recurrent storm impact across levels of 1) economic segregation, 2) residential segregation, 3) urbanity, and 4) access to maternal care.**

| | ICE Economic Segregation | | | | | | | | | | | | p-EM |
|---|---|---|---|---|---|---|---|---|---|---|---|---|---|
| **Outcomes** | *Majority low-income (T1)* | | | | *Mixed economic composition (T2)* | | | | *Majority high-income (T3)* | | | | |
| | 1 Storm | | ≥2 Storms | | 1 Storm | | ≥2 Storms | | 1 Storm | | ≥2 Storms | | |
| | aRR | 95%CI | aRR | 95%CI | aRR | 95%CI | aRR | 95%CI | aRR | 95%CI | aRR | 95%CI | |
| MDP | 1.07 | (0.97, 1.17) | 1.16 | (1.05, 1.28) | 0.68 | (0.60, 0.77) | 1.63 | (1.52, 1.76) | 1.13 | (1.03, 1.23) | 2.22 | (1.97, 2.49) | < .0001 |
| PMAD | 1.13 | (1.04, 1.22) | 1.32 | (1.21, 1.43) | 0.58 | (0.54, 0.65) | 1.51 | (1.41, 1.62) | 1.01 | (0.93, 1.10) | 1.79 | (1.61, 1.99) | < .0001 |
| SMI | 1.04 | (0.90, 1.20) | 1.21 | (1.04, 1.41) | 0.73 | (0.60, 0.89) | 1.51 | (1.32, 1.73) | 1.05 | (0.92, 1.20) | 1.24 | (1.02, 1.50) | 0.0004 |
| SUDP | 1.37 | (1.21, 1.56) | 1.32 | (1.14, 1.53) | 1.06 | (0.89, 1.26) | 0.78 | (0.67, 0.90) | 0.96 | (0.84, 1.10) | 0.58 | (0.45, 0.76) | < .0001 |

| | ICE Residential Segregation | | | | | | | | | | | | p-EM |
|---|---|---|---|---|---|---|---|---|---|---|---|---|---|
| **Outcomes** | *Majority Black (T1)* | | | | *Mixed racial composition (T2)* | | | | *Majority White (T3)* | | | | |
| | 1 Storm | | ≥2 Storms | | 1 Storm | | ≥2 Storms | | 1 Storm | | ≥2 Storms | | |
| | aRR | 95%CI | aRR | 95%CI | aRR | 95%CI | aRR | 95%CI | aRR | 95%CI | aRR | 95%CI | |
| MDP | 0.51 | (0.46, 0.57) | 0.63 | (0.53, 0.74) | 0.99 | (0.91, 1.08) | 0.88 | (0.80, 0.98) | 1.68 | (1.53, 1.83) | 2.55 | (2.38, 2.74) | < .0001 |
| PMAD | 0.61 | (0.55, 0.66) | 0.81 | (0.70, 0.94) | 0.83 | (0.77, 0.90) | 0.97 | (0.89, 1.06) | 1.57 | (1.45, 1.69) | 2.21 | (2.07, 2.36) | < .0001 |
| SMI | 0.75 | (0.65, 0.88) | 0.94 | (0.74, 1.20) | 0.82 | (0.72, 0.94) | 0.97 | (0.83, 1.13) | 1.50 | (1.31, 1.73) | 1.75 | (1.55, 1.97) | < .0001 |
| SUDP | 1.37 | (1.20, 1.56) | 1.53 | (1.27, 1.85) | 1.04 | (0.91, 1.20) | 1.07 | (0.91, 1.27) | 0.99 | (0.86, 1.16) | 0.70 | (0.61, 0.81) | < .0001 |

| | Urbanity | | | | | | | | | | | | p-EM |
|---|---|---|---|---|---|---|---|---|---|---|---|---|---|
| **Outcome** | *Urban* | | | | *Suburban* | | | | *Rural* | | | | |
| | 1 Storm | | ≥2 Storms | | 1 Storm | | ≥2 Storms | | 1 Storm | | ≥2 Storms | | |
| | aRR | 95%CI | aRR | 95%CI | aRR | 95%CI | aRR | 95%CI | aRR | 95%CI | aRR | 95%CI | |
| MDP | 0.97 | (0.91, 1.02) | 1.77 | (1.67, 1.88) | 1.62 | (1.39, 1.89) | 1.03 | (0.90, 1.18) | 1.08 | (0.84, 1.39) | 0.42 | (0.19, 0.96) | < .0001 |
| PMAD | 0.90 | (0.85, 0.94) | 1.67 | (1.59, 1.76) | 1.56 | (1.36, 1.79) | 1.06 | (0.94, 1.18) | 1.27 | (1.03, 1.58) | 0.66 | (0.36, 1.21) | < .0001 |
| SMI | 0.93 | (0.85, 1.01) | 1.56 | (1.42, 1.72) | 1.63 | (1.31, 2.04) | 0.82 | (0.67, 1.01) | 0.79 | (0.48, 1.32) | 0.66 | (0.21, 2.10) | < .0001 |
| SUDP | 1.11 | (1.02, 1.20) | 0.89 | (0.80, 1.00) | 1.20 | (0.96, 1.50) | 1.08 | (0.91, 1.29) | 0.83 | (0.57, 1.21) | 0.60 | (0.22, 1.65) | 0.1788 |

| | Access to Care | | | | | | | | | | | | p-EM |
|---|---|---|---|---|---|---|---|---|---|---|---|---|---|
| **Outcomes** | *Full access* | | | | *Low Access* | | | | *Maternity Care Desert* | | | | |
| | 1 Storm | | ≥2 Storms | | 1 Storm | | ≥2 Storms | | 1 Storm | | ≥2 Storms | | |
| | aRR | 95%CI | aRR | 95%CI | aRR | 95%CI | aRR | 95%CI | aRR | 95%CI | aRR | 95%CI | |
| MDP | 1.10 | (1.04, 1.17) | 1.36 | (1.28, 1.46) | 1.03 | (0.91, 1.17) | 2.63 | (2.36, 2.93) | 1.04 | (0.81, 1.35) | 0.50 | (0.34, 0.72) | < .0001 |
| PMAD | 1.03 | (0.98, 1.09) | 1.25 | (1.18, 1.33) | 0.98 | (0.88, 1.09) | 2.57 | (2.34, 2.81) | 1.13 | (0.90, 1.42) | 0.64 | (0.47, 0.86) | < .0001 |
| SMI | 1.00 | (0.91, 1.12) | 1.20 | (1.07, 1.33) | 1.22 | (1.02, 1.45) | 2.03 | (1.72, 2.39) | 0.60 | (0.35, 1.00) | 0.34 | (0.18, 0.66) | < .0001 |
| SUDP | 1.05 | (0.96, 1.14) | 0.89 | (0.80, 0.99) | 1.49 | (0.96, 1.50) | 1.08 | (0.91, 1.29) | 0.81 | (0.51, 1.29) | 0.95 | (0.59, 1.54) | 0.0041 |

MDP: Maternal mental disorder during pregnancy; PMAD: Perinatal mood or anxiety disorder: SMI: Severe mental disorder: SUBP: Substance use disorder complicating pregnancy.

*referent = 0 storms.

[a]Models adjust for individual and community-level characteristics (i.e., racial/ethnic identity, maternal age, access to care, urban/rural status, ICE: economic segregation, and ICE: residential segregation)

individual storms showed that Hurricane Matthew had the most significant impact on new mental disorders during pregnancy, including perinatal mood and anxiety disorders (PMAD) and severe mental illness (SMI). Florence resulted in an increase in new mental disorders diagnosed during pregnancy (MDP) and substance misuse during pregnancy (SUDP), although not statistically significant. Conversely, Dorian was associated with a significant reduction in ED visits for maternal mental disorders, particularly for MDP, SMI, and PMAD in pregnant Medicaid beneficiaries. For the recurrent storm models, we observed a 158% increase in MDP, 151% increase in PMAD-related, and 135% increase in SMI-related ED visits for pregnant Medicaid beneficiaries impacted by two or more storms. For all other maternal mental health

outcomes, with the exception of a slight increase in SUDP, we noted a significant decline in ED visits for counties impacted by one storm. Results from this population-based analysis reveal the cumulative risks of recurrent hurricane exposure and more cases of psychiatric morbidity during pregnancy, including incident mental disorders, mood and anxiety disorders, and severe mental illness for pregnant Medicaid beneficiaries residing in communities impacted by two or more major hurricanes.

Our findings demonstrate the cumulative mental health burden of multiple and recurring hurricane exposure on pregnant populations. Similarly, prior research has shown an increased risk in response to multiple compared to single disaster events for self-reported mood, anxiety, or substance use disorders [40]. The majority of the literature has cited increases in self-reported mental health [15, 41–44] in adult populations following multiple disaster events, but too few have examined sex-based differences, particularly for females of reproductive age [45, 46]. Elevated symptoms of depression and PTSD were noted among women of reproductive age following the Gulf oil spill and hurricanes Katrina, Rita, Gustav, and Ike, substantiating the likelihood of cumulative mental health risk following multiple disasters [46], but other outcomes like anxiety, severe mental illness and substance misuse were not evaluated. Another study demonstrated improved mental health post-Gustav, but worse overall mental health in postpartum women with small children for those who experienced both Hurricanes Gustav and Katrina, especially for older women who had low social support and high levels of major stressors [45].

There are multiple proposed mechanisms linking recurrent disaster exposure to poor mental health, including direct exposure to the disaster event, sleep disturbances, and reduced access to the health care system, including mental health support [6, 47]. Biologic mechanisms that might be involved include allostatic load, epigenetic changes, and altered or dysregulation in neurobiological pathways that mediate stress [48–50]. Recurrent or multiple disaster experiences have significant potential to threaten perceived safety, security, and hope in the future for directly exposed populations, particularly among women [51]. Further, recurrent disasters may exacerbate existing social and economic stressors, especially in low-income or communities of color who have experienced persistent disinvestment [52]. Research in adults shows that prior disaster exposure and the associated mental health sequelae might also confer an increase in mental health vulnerability for a subsequent disaster experience [53, 54]. To our knowledge, no studies have examined the association between multiple hurricane exposure and the worsening of maternal mental health during pregnancy.

While we observed that a single hurricane event was not associated with an increased risk of psychological morbidity during pregnancy, with the exception of elevated risk of substance use, very few studies have established a connection between direct disaster exposure and increased substance use, as a coping mechanism for disaster-related stress, in pregnant populations [55]. However, some literature has drawn an important connection between a mirrored increase in intimate partner violence and substance use for women who have experienced a natural disaster [56]. The majority of the disaster literature has not uncovered a substantial increase in substance use disorders post-event [40, 57]. While it may be that substance use serves as a proxy for intimate partner violence, an issue that many providers have limited experience with [58], more research is needed to explore the linkages between substance use, disasters, and intimate partner violence.

Women residing in counties with low access to maternal care services and that experienced two or more storms were over 200% more likely to experience an MDP, PMAD, and SMI-related ED visits. We also noted that pregnant Medicaid beneficiaries residing in counties characterized by low access to care were disproportionately impacted by 2 or more recurrent hurricanes. Prior research has cited access to care and maternal stress as leading challenges post-hurricane, including for maternal and child health programs in Puerto Rico after Hurricanes Irma and Maria [59]. Findings showed that pregnant persons residing in a maternity

care desert impacted by two or more storms were significantly less likely to present to the ED for an emergency maternal mental disorder. Research after Hurricane Michael on access to care for pregnant women revealed delayed access to care in areas most affected by the storm [20], and this delay may have resulted in unaddressed or interrupted care for mental health needs in our sample of low-income pregnant women.

Interestingly, results revealing higher ED usage for maternal mental health disorders in predominantly White communities and a reduced occurrence of ED visits for mental health in Majority-Black communities impacted by consecutive hurricane events are contrary to what we expected. Our findings are corroborated by existing research showing a generally lower lifetime prevalence of mental disorders, including depressive, anxiety, and substance disorders, and a lower prevalence of natural disaster exposure in Black, Latinx, and Asian compared to White adults [60]. In a study examining trauma-related risk factors among a subset of pregnant African American women attending an obstetric/gynecological clinic, Powers et al. (2020) found that despite a high level (98%) of trauma experiences within the sample population (n = 633), with 30% meeting the DSM-IV-TR criteria for PTSD, only 6.2% engaged in behavioral health treatment. Low engagement with behavioral health treatment among African Americans may be associated with missed opportunities for screening of maternal mental disorders within this population [61, 62]. However, social and economic barriers to care may also be present including costs associated with care (e.g., lack of paid time off work), competing demands of childrearing, transportation, availability of services, racial discrimination and other chronic socio-environmental stressors [63–65]. When layered onto everyday mental health stressors, recurrent disaster exposures may serve to increase traumatic events within this population, with no historical mechanisms in which to seek adequate treatment to alleviate potential excessive mental health burdens. More data-driven research on recurrent disaster impacts in low-income and minoritized communities is needed to quantify and critically examine the driving force of structural inequalities in perpetuating climate-imposed health disparities in mental health, particularly following recurrent climate disasters.

Following Dorian, we likely observed a lower risk of presenting to the ED with a maternal mental health disorder because our post-hurricane study period eclipsed the early part of the COVID-19 pandemic ending just before the December 2020 to January 2021 peak in national COVID-19 cases, a time characterized by a dramatic decline in care-seeking behaviors for mental health disorders [66]. National shutdowns and stringent social distancing measures exacerbated mental distress during the early part of the pandemic and, alongside fear of infection and rapid changes in maternity care services were important deterrents to seeking medical care [67]. While the post-pandemic literature is evolving, more recent studies have demonstrated a significant increase in mental distress, anxiety, depression, and other adverse mental health sequelae during pregnancy as the pandemic progressed [68–71].

## Strengths/limitations

This is one of the few studies to examine the mental health consequences of multiple disaster exposure in a highly vulnerable population, pregnant Medicaid beneficiaries. Unlike the majority of the previous literature, we did not examine self-reported mental health outcomes, but relied on medical diagnosis codes for an array of mental health disorders. We also included a representative population of pregnant persons for the entire state and were able to differentiate between communities exposed and unexposed using a meteorological-based storm exposure metric. Lastly, we expanded our analysis of psychiatric morbidity outcomes beyond posttraumatic stress disorder, an outcome typically included in post-disaster studies, to include mental disorders more common during pregnancy.

Results should be interpreted with some caution due to the following limitations. Given the cross-sectional nature of our ED data, we were unable to longitudinally track pregnant persons over time, but we were able to assess ecological changes in ED utilization for each county over time. The DID analytic approach did lend to causal inference byway of the quasi-experimental design to compare maternal mental health burden in exposed versus unexposed populations before and after each hurricane. We acknowledge that external factors such as other traumatic events (e.g., mass shootings, domestic violence, death of a loved one, prior hurricane experience, etc.), diminished availability of qualified providers, health services closures, changes in health-seeking behaviors or maternal residence, and Medicaid coverage fluctuations can influence the trends identified. However, given the limitations of hospital administrative data related to ED visits, many of these factors cannot be assessed within the context of this study. An additional limitation is that storm exposure was approximated using wind speed and total precipitation at the county-level, and data on individual-level direct experience or varying levels of storm severity across a county were not captured.

In the future, studies employing a mixed-method approach that contextualizes quantitative findings from longitudinal cohorts with qualitative data detailing the actual experiences and perspectives of women will shed light on the total mental health burden of recurrent storms and obviate the potential for reverse causation. While robust methodologies are still developing on identifying the impact of multiple disasters on populations at higher risk, qualitative methods such as in-depth interviews, focus group discussion, and oral histories may be a path forward towards understanding the complex weathering effects of varying disasters on women. Such studies may also shed light on mechanisms that promote needed resilience and recovery measures for women in communities historically excluded from treatment and recovery policies.

## Study implications

The study presents important implications demonstrating that within the context of recurrent disasters, the excess burden of ED visits for mental health illnesses increases in pregnant Medicaid women. However, interesting dynamics within the results present when assessed through the lens of additional risk factors that warrant further investigation. For instance, low and full maternal care access communities had increased ED visits following multiple storms; yet ED visits for mental disorders decreased for pregnant populations residing in maternal care desert communities. The pattern was similar for majority white compared to predominantly African American communities. The reduced utilization of ED for emergency psychiatric illnesses post-recurrent storm exposure occurred over a baseline of lower ED utilization within these low-resource minority communities. Such a decline in ED visits compared to other communities may indicate the lack of acquiring needed emergency assistance more broadly, but may also point to potential structural barriers which such communities may need to contend with every day, including a growing shortage of providers, transportation barriers, and provider bias; factors that do not cease to exist when pregnant persons within these communities are faced with natural disasters and the pursuit of needed physical and economic recovery assistance.

Broadly, a better understanding of the mental health care landscape and access for Medicaid pregnant and postpartum mothers is needed to understand how this population may weather recurrent disaster exposures. Policy scans of state Medicaid program waivers after disasters may assist in providing a broader understanding of access to mental health services. According to the 2023 Mental Health in America Report, North Carolina ranked a low 39 out of 51 states and Washington DC on access to insurance, treatment, costs, and mental health workforce availability [72], indicating a significant unmet mental health need in the state. Needed research should focus on the implementation and evaluation of telehealth services in mental

health care deserts for Medicaid mothers impacted by recurrent disasters and the need for increased psychosocial support and coping skills.

In addition to further studies that seek to elucidate the mental health care landscape of pregnant and postpartum women impacted by recurrent disasters, policy implications are also present. The study does not examine the role of disaster recovery services through federal and state assistance in mitigating the impacts of disasters on individuals and communities. The Federal Emergency Management Agency (FEMA) issues recovery grants to individuals through the Individuals and Households Programs [73]. The agency, as a policy directive, also provides special accommodations to individuals with access and functional needs, including disaster assistance application support. Increasing how well pregnant and postpartum mothers navigate special accommodations for disaster assistance, and the use of disaster assistance to alleviate recovery-induced stress and mitigate excess psychiatric burden in this population is imperative. Moreover, FEMA provides state and local governments with training support for crisis counseling and assistance, allowing such impacted jurisdictions to administer community-based outreach and psycho-educational services [73]. More work in understanding how to increase timely and supportive disaster mental health program access, while elevating culturally-relevant means of resilience is crucial towards reducing risks of psychiatric illnesses in the face of recurrent disasters.

## Conclusion

The public health impacts of recurrent or multiple disaster exposures in the same geographic area is an emerging area of research. Our population-based findings demonstrate that cumulative hurricane exposure (i.e., two or more storms) confers an increased risk for psychological morbidity during pregnancy, particularly for mood and anxiety disorders, newly diagnosed maternal mental disorders, and severe mental illness for a Southern state outside of the U.S. Gulf Coast. The wide-ranging mental health consequences of a single disaster have been widely documented, but important gaps remain in understanding the mental health burden of recurrent exposure to climate disasters, like large-scale hurricanes. More importantly, more research is needed to understand differential mental health vulnerability for populations of concern, like pregnant populations and their children, as the frequency and severity of climate-intensified hurricanes are expected to worsen throughout the 21st century. The integration of enhanced clinical care and prenatal mental health screening will be integral for ensuring at-risk pregnant populations receive the care they need.

## Supporting information

**S1 Table. International Classification of Diseases, tenth revision, Clinical Modification (ICD-10-CM) diagnosis codes used to operationalize major and sub-categories of maternal mental disorders.**
(DOCX)

**S2 Table. Hurricane exposure status of a retrospective cohort study of eligible pregnant Medicaid beneficiaries among North Carolina residents, 2015–2020.**
(DOCX)

## Author Contributions

**Conceptualization:** Jennifer D. Runkle.

**Data curation:** Trey Williamson.

**Formal analysis:** Kelsey Herbst, Trey Williamson, Jennifer D. Runkle.

**Funding acquisition:** Jennifer D. Runkle.

**Methodology:** Sudeshna Paul, Carl J. Schreck, Jennifer D. Runkle.

**Supervision:** Jennifer D. Runkle.

**Visualization:** Kelsey Herbst, Jennifer D. Runkle.

**Writing – original draft:** Kelsey Herbst, Natasha P. Malmin, Sudeshna Paul, Trey Williamson, Margaret M. Sugg, Carl J. Schreck, Jennifer D. Runkle.

**Writing – review & editing:** Kelsey Herbst, Natasha P. Malmin, Sudeshna Paul, Trey Williamson, Margaret M. Sugg, Carl J. Schreck, Jennifer D. Runkle.

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
