## [Decision Letter · Decision Letter 0]

12 Mar 2024

PMEN-D-24-00067

Examining recurrent hurricane exposure and psychiatric morbidity in Medicaid-insured pregnant populations

PLOS Mental Health

Dear Dr. Runkle,

Thank you for submitting your manuscript to PLOS Mental Health. After careful consideration, we feel that it has merit but does not fully meet PLOS Mental Health’s publication criteria as it currently stands. Therefore, we invite you to submit a revised version of the manuscript that addresses the points raised during the review process.

We look forward to receiving your revised manuscript.

Kind regards,

Paolo Raile

Academic Editor

PLOS Mental Health

Journal Requirements:

1. Please amend your detailed online Financial Disclosure statement. This is published with the article. It must therefore be completed in full sentences and contain the exact wording you wish to be published.

a) State the initials, alongside each funding source, of each author to receive each grant. For example: "This work was supported by the National Institutes of Health (####### to AM; ###### to CJ) and the National Science Foundation (###### to AM)."

2. Please update your online Competing Interests statement. If you have no competing interests to declare, please state: “The authors have declared that no competing interests exist.”

3. Please provide separate figure files in .tif or .eps format only and ensure that all files are under our size limit of 10MB.

For more information about how to convert your figure files please see our guidelines: https://journals.plos.org/complexsystems/s/figures

4. Please include a separate legend for Figure 1 in your manuscript.

5. We have noticed that you have uploaded Supporting Information files, but you have not included a list of legends. Please add a full list of legends for your Supporting Information files after the references list.

Additional Editor Comments (if provided):

Both reviewers asked important substantive questions that need to be answered in a revision. Please revise the manuscript to meet the comments and questions of the reviewer and resubmit the paper.

Reviewers' comments:

Reviewer's Responses to Questions

**Comments to the Author**

1. Does this manuscript meet PLOS Mental Health’s publication criteria? Is the manuscript technically sound, and do the data support the conclusions? The manuscript must describe methodologically and ethically rigorous research with conclusions that are appropriately drawn based on the data presented.

Reviewer #1: Partly

Reviewer #2: Yes

2. Has the statistical analysis been performed appropriately and rigorously?

Reviewer #1: I don't know

Reviewer #2: Yes

3. Have the authors made all data underlying the findings in their manuscript fully available (please refer to the Data Availability Statement at the start of the manuscript PDF file)?

Reviewer #1: Yes

Reviewer #2: Yes

4. Is the manuscript presented in an intelligible fashion and written in standard English?

Reviewer #1: Yes

Reviewer #2: Yes

5. Review Comments to the Author

Reviewer #1: The manuscript is an analysis of mental health claims in the North Carolina Medicaid data and hurricane exposure.

In a few places it is unclear whether the comparison is non-pregnant women or pregnant women in unaffected communities.

I am not very familiar with the DiD method, but I’m probably not the only one in your readership. I tend to think of it as being conducted at an ecologic level, but some of the description sounds more like an individual level, i.e., each person in the dataset is coded as yes/no for having this outcome. Are you modeling the number of diagnoses per county, the proportion of pregnant ED admissions with a mental health code, the number of MH diagnoses per 1000 births or pregnancies?

Given that diagnosis of a mental health issue among pregnant Medicaid patients in an ED relies on several levels of conditions, it would be helpful to explain why the trends can be attributed to the hurricanes and not other factors, which might vary by hurricane status.

- Providers may be more likely to diagnose mental health issues after a disaster

- There may be fewer qualified providers after a disaster

- People may use EDs more if primary care clinics are shut down. Similarly, the higher risk in the maternity care deserts may indicate lack of other care options, not a higher rate of the outcome.

- EDs may be closed due to the disaster

- People may be dealing with disaster aftermath and be less likely to seek medical care for less serious problems

- Medicaid coverage may be extended or reduced (most pregnant women are eligible, but eligibility criteria and requirements are often changed after a disaster)

- The overall population in the county may have changed

- The comparison group is not always clear to me. If someone does not have the outcome, are they are pregnant case visiting the ED for another reason? If so, couldn’t changes in other outcomes also change the relative proportion visiting for pregnancy/MH?

The method of defining storm exposure is interesting. How much does it vary from other options, like disaster declarations or wind speed? Since it’s based on crop loss, is there any possibility that the level of agricultural activity in a county would cause variation in exposure definition.

Why are maternity care deserts coded as 3 and low access as 1?

This sentence doesn’t make sense: “We opted to categorize recurrent storm exposure this way, because there were such a small number of individuals who experienced two recurrent storms.” Do they mean 2 or more?

The definition of recurrent exposure is confusing. My understanding is that it is just whether a woman goes to an ED in that county, and the county’s level of hurricane is the exposure, not the individual’s. Since individuals are not linked over time, presumably you don’t know when someone moved into the county? Was gestational age used to determine whether the storm had occurred during the pregnancy?

Substance use as a proxy for IPV seems like a stretch; it may be sometimes be the case, but people also use substances to cope with stress and that seems like a more direct hypothesis.

The authors say they can’t take into account mass shootings and the like, but it seems like those could be incorporated into the analysis, if they wished.

Reviewer #2: This article uses indirect sources to make inferences (Medicade records, federal funding in the counties etc.)about mental health of pregnant women who are exposed to multiple disasters. Overall the measures are analyzed and articulated appropriately. The many limitations to using these indirect sources was well described.

Unclear areas that could be made further explicit in the study, if possible, are: (1) What was the degree of danger women in these areas were at risk for as the respective storms passed though eastern NC? What was downgraded intensity and degree of flooding and infrastructure impact? There are degrees of "exposure" if you can describe more. How long did the hurricane systems affect the targeted affected communities?

For multiple exposures, women were only pregnant for one of the exposures, right, or both? Were they pregnant when two different systems went through respective dates? That was not clear.

6. PLOS authors have the option to publish the peer review history of their article (what does this mean?). If published, this will include your full peer review and any attached files.

**Do you want your identity to be public for this peer review?** For information about this choice, including consent withdrawal, please see our Privacy Policy.

Reviewer #1: No

Reviewer #2: No

---

## [Decision Letter · Decision Letter 1]

7 May 2024

Examining recurrent hurricane exposure and psychiatric morbidity in Medicaid-insured pregnant populations

PMEN-D-24-00067R1

Dear Dr. Runkle,

We are pleased to inform you that your manuscript 'Examining recurrent hurricane exposure and psychiatric morbidity in Medicaid-insured pregnant populations' has been provisionally accepted for publication in PLOS Mental Health.

Best regards,

Paolo Raile

Academic Editor

PLOS Mental Health

Reviewer Comments (if any, and for reference):

Reviewer's Responses to Questions

**Comments to the Author**

1. If the authors have adequately addressed your comments raised in a previous round of review and you feel that this manuscript is now acceptable for publication, you may indicate that here to bypass the “Comments to the Author” section, enter your conflict of interest statement in the “Confidential to Editor” section, and submit your "Accept" recommendation.

Reviewer #1: All comments have been addressed

Reviewer #2: All comments have been addressed

2. Does this manuscript meet PLOS Mental Health’s publication criteria? Is the manuscript technically sound, and do the data support the conclusions? The manuscript must describe methodologically and ethically rigorous research with conclusions that are appropriately drawn based on the data presented.

Reviewer #1: Yes

Reviewer #2: Yes

3. Has the statistical analysis been performed appropriately and rigorously?

Reviewer #1: Yes

Reviewer #2: Yes

4. Have the authors made all data underlying the findings in their manuscript fully available (please refer to the Data Availability Statement at the start of the manuscript PDF file)?

Reviewer #1: No

Reviewer #2: Yes

5. Is the manuscript presented in an intelligible fashion and written in standard English?

Reviewer #1: Yes

Reviewer #2: Yes

6. Review Comments to the Author

Reviewer #1: The authors have addressed reviewer comments and the paper is much clearer.

Reviewer #2: Clarity to the methods section was made by noting "county" as the primary source of study. Limitatons section was updated to indicate the lack of data concerning women per se going through multiple events.

7. PLOS authors have the option to publish the peer review history of their article (what does this mean?). If published, this will include your full peer review and any attached files.

**Do you want your identity to be public for this peer review?** For information about this choice, including consent withdrawal, please see our Privacy Policy.

Reviewer #1: No

Reviewer #2: No

Please also include ethics statements/approvals in the Methods Section